# Bridging the Gap Between Platforms: Comparing Grape Phylloxera *Daktulosphaira vitifoliae* (Fitch) Microsatellite Allele Size and DNA Sequence Variation

**DOI:** 10.3390/insects16020230

**Published:** 2025-02-19

**Authors:** Mark J. Blacket, Alexander M. Piper, Ary A. Hoffmann, John Paul Cunningham, Isabel Valenzuela

**Affiliations:** 1Agriculture Victoria Research, AgriBio Centre for AgriBioscience, Bundoora, Melbourne, VIC 3083, Australia; alexander.piper@agriculture.vic.gov.au (A.M.P.); paul.cunningham@agriculture.vic.gov.au (J.P.C.); isabel.valenzuela-gonzalez@agriculture.vic.gov.au (I.V.); 2School of BioSciences, Bio21 Institute, University of Melbourne, Melbourne, VIC 3010, Australia; ary@unimelb.edu.au; 3Department of Science, Health and Engineering, School of Applied Systems Biology, La Trobe University, Bundoora, Melbourne, VIC 3083, Australia

**Keywords:** grape phylloxera, microsatellite—SSR markers, high-throughput sequencing, capillary genotyping, allele size calibration, molecular diagnostics, regulated pest, biosecurity

## Abstract

Grape phylloxera is a serious insect pest of grapevines worldwide. The past two decades have seen genotypic identification of phylloxera using microsatellite markers become an integral part of control, informing the selection of resistant rootstocks and implementation of quarantine zones to prevent the spread of highly virulent genotypes. Here, we assessed three different molecular methods for screening phylloxera microsatellites, providing comparisons with previous data using newer laboratory approaches, including phylloxera whole-genome DNA sequence data. These comparisons and the standard laboratory protocols presented will allow molecular diagnostic results to be consistently obtained between different research groups and maintain compatibility of future work with valuable historic datasets.

## 1. Introduction

Grape phylloxera, *Daktulosphaira vitifoliae* (Fitch) (Hemiptera: Phylloxeridae), are small yellow sap-sucking insects related to aphids that feed on the leaves and roots of grapevines. Originally native to North America, phylloxera has spread to all major grapevine growing regions of the world [1], where it has become a significant insect pest of commercial grape production [2]. While in its native range, phylloxera predominantly occurs in leaf-galling forms on wild grapevine species, on *Vitis vinifera* L. (European grape vine), phylloxera causes damage to leaves (galls) and roots (nodules or tuberosities). Leaf feeding on *V. vinifera* results in reduced vine productivity, while root feeding can lead to vine death [2]. Diagnostic laboratory identification of this serious pest species is currently achieved through visual microscopic examination [3,4], or by molecular assays, including qPCR [5,6], DNA barcoding and LAMP [7].

In its native range, sexual reproduction is common, but outside the Americas, the main mode of reproduction of phylloxera appears to be parthenogenetic, i.e., asexual/clonal [8,9,10,11]. Surveillance, quarantine of infected areas, and the grafting of *V. vinifera* onto resistant rootstocks derived from native American *Vitis* species are currently the primary forms of control [2,12]. However, as different parthenogenetic lineages vary greatly in their virulence and susceptibility to control through rootstocks [13,14,15], identification of clonal diversity through molecular testing (as outlined below) forms a major aspect of management [10,16].

Various molecular markers have previously been employed to examine phylloxera genetic diversity, including RAPDs (Random Amplification of Polymorphic DNA) [17,18], AFLPs (Amplified Fragment Length Polymorphisms) [19], and mitochondrial DNA sequences [7,10,20,21,22]. Over the last two decades, numerous microsatellite, or Simple Sequence Repeat (SSR), markers have been developed for phylloxera [8,9,23,24,25]. SSR markers are currently the preferred molecular method for genotype identification, with their widespread use reviewed by Tello and Forneck (2019) [1]. While alternative modern molecular approaches are now available, such as high-throughput genome sequencing [26], SSR markers are still critical for phylloxera identification, as historically, clonal lineages of phylloxera have been defined based the combinations of SSR alleles they possess (i.e., to define them as a genotype). These combinations have been used to identify genotypes in all subsequent virulence/resistance research [1].

Methods for assigning SSR genotypes have switched over the past two decades from visual or partially automated inspections of PCR products on polyacrylamide gels [8,9,23], to more automated assessments of fluorescently labelled fragments through capillary electrophoresis [27,28]. This capillary technology has several advantages [29], including being highly suitable for PCR multiplexing, which involves multiple loci being amplified in a single PCR reaction to greatly reduce time and reagent costs. This has resulted in broad adoption throughout the world in population genetic studies [25,27,28,30,31]. However, several issues arise from switching to a new genotyping platform, the most significant being that previously generated data may be incompatible with new data if the sizes of alleles change due to different scoring methods [32,33]. This potentially can be resolved by creating standard allele sets as controls and using these for adjusting allele sizes to match data generated by previous genotyping technologies: this allows congruent biological information to be maintained between studies [32,34], which is critically important for ongoing phylloxera pest management strategies.

The primary aims of the current study were to (1) optimise laboratory methods for capillary-based genotyping a set of phylloxera SSR markers using fluorescently labelled universal primer multiplex PCR (*sensu* Blacket et al. [35]); (2) compare allele sizes generated using three alternative genotyping platforms (polyacrylamide gels, capillary separation and HTS genotyping); and (3) recommend a ‘standard’ set of SSRs and phylloxera clonal genotypes to link new data with previous studies.

## 2. Materials and Methods

### 2.1. Phylloxera Samples and DNA Extractions

Specimens of seven clonal phylloxera lineages—G1, G4, G7, G19, G20, G30 and G38—originally collected from Australian vineyards and identified as per Umina et al. [10], were obtained from long-term Agriculture Victoria laboratory colonies maintained on *Vitis vinifera* cv. Chardonnay roots, at Rutherglen, Australia. Individual specimens from each colony were stored at −20 °C in ~100% ethanol prior to DNA extraction. Specimens were removed from ethanol using a fine hairbrush and air-dried on tissue paper for one minute. DNA was extracted using Chelex® (Bio-Rad Laboratories, Inc., Hercules, CA, USA), with specimens ground in a 1.5 mL Eppendorf tube at 30,000 Hz for one minute, using a bead mill homogeniser (85210 TissueLyser, Qiagen, Hilden, Germany), containing two 3 mm glass beads, 100 µL of 5% Chelex, and 3 µL of Proteinase K (20 mg/mL, Qiagen, Hilden, Germany). The homogenised specimens were incubated at 56 °C for 60 min, and 90 °C for 8 min, then cooled on ice for at least one hour before PCR (or stored in a −20 °C freezer overnight). Prior to PCR, the samples were spun at 13,000 rpm for two minutes in a microcentrifuge, with 2 µL of template DNA pipetted from the supernatant and used for each PCR reaction. For long-term storage, the samples were preserved frozen at −20 °C in 1.5 mL Eppendorf tubes. Additionally, a small number of archived DNA extractions from a previous study [10] were available for use in this study (see below).

### 2.2. Selection of a Standard Set of Phylloxera SSR Loci

A set of eight widely used SSR markers [1] were chosen as standard loci for this study. All previously published studies of Australian phylloxera SSR variation have utilised markers Dvit1 to Dvit4 [8,22,36], and Dvit1 to Dvit6 [10]. These loci show high levels of allelic variation with a relatively large number of alleles (e.g., up to seven alleles for Dvit5), with a wide range of allele sizes, between 126 and 289 base pairs (bp) [10], which is essential for accurately differentiating alleles from multiplexed PCR reactions. Two additional markers, DVSSR 3 and DVSSR4, were added to increase the number of standard markers for use in Australia. These additional SSR loci were obtained from Lin et al. [23] and were selected based on their high allelic variability and suitability for multiplexing with the existing set of six Dvit loci (Appendix A).

### 2.3. Platform 1: (Original) Polyacrylamide Genotyping

Archived DNA samples (i.e., not insect specimens) of Australian phylloxera genotypes G1 and G4 were available from previous work [10]. These DNA extractions had been previously characterised using a polyacrylamide gel system and identified as phylloxera genotypes G1 and G4, based on Dvit1 to Dvit6 loci (*sensu* Umina et al. [10]). These archived DNA extractions had been preserved frozen at −20 °C in 1.5 mL Eppendorf tubes with the lid wrapped in parafilm to avoid desiccation of the DNA extracts. Unfortunately, no other samples had been preserved, making these samples the only ‘historical’ samples available to us for direct genotyping platform comparison studies. Expected allele sizes for additional genotypes included in the current study (i.e., G7, G19, G20, G30 and G38) were obtained from Umina et al. [10].

### 2.4. Platform 2: Fluorescent Capillary Genotyping

The optimised capillary genotyping laboratory methods are briefly outlined here, with additional laboratory protocol details provided as Appendix A (Appendix A). Loci were amplified using tailed versions of previously published primers, together with fluorescently labelled universal primers [35], in one single and three multiplex 10 µL PCR reactions (see below, and Appendix A). PCRs were caried out using Eppendorf epgradient S or Applied Biosystems Veriti 96-well thermal cyclers. Initially, gradient PCRs (45–60 °C) were performed on all loci to determine the optimal annealing temperature required to produce strong amplification and avoid null alleles [37], prior to multiplexing of primers. Multiplexing trials involved testing sets of primers with similar matching annealing temperatures and adjusting primer ratios by up to 2:1 for loci which were found to weakly amplify in multiplex reactions. Subsets of amplicons were visually checked for successful amplification on 3% agarose gels. Finally, 1 µL of each PCR was then diluted 1:20 with dH2O in preparation for genotyping. PCR 1 and 2 (FAM and PET) were poolplexed (i.e., mixed post PCR), as were PCR 3 and 4 (VIC and NED), to reduce the genotyping of all eight loci down to two capillary lanes for each sample. Final PCR conditions involved four master mix reactions: (MM1) Dvit4/Dvit6/DVSSR4 (annealing 45 °C); (MM2) Dvit1/DVSSR3 (annealing 47 °C); (MM3) Dvit5 (annealing 49 °C); and (MM4) Dvit2/Dvit3 (annealing 57 °C). These four PCR amplifications can be performed simultaneously in a single-gradient 96-well PCR machine if the sample numbers are low (e.g., <16). Optimisation of many of the loci with larger PCR fragment sizes (Dvit2, Dvit6 and DVSSR3) revealed that they required double the amount of locus-specific primer to be used to avoid preferential amplification of smaller sized loci in multiplex reactions (see Appendix A).

Capillary separation was performed commercially (through AGRF Melbourne) on an Applied Biosystems ABI 3730 DNA analyser, with genotyping conducted in Geneious R11 (Biomatters Ltd., Auckland, New Zealand), using the LIZ500_3730 size standard, with the 250 bp fragment removed from analyses, as per the manufacturer’s recommendations. The archived DNA extractions served as controls of known allele sizes and were used for defining allele size BINs in Geneious (see Appendix A).

### 2.5. Platform 3: Genotyping by High-Throughput Sequencing (HTS)

Genomic DNA was extracted from two replicate samples of each strain, each containing between 2 and 10 pooled individuals, using the Qiagen DNeasy blood and tissue kit (Qiagen, Hilden, Germany). The extracted DNA was enzymatically sheared using the method described by Shinozuka et al. [38] and sequencing libraries prepared using the JetSeq Flex DNA library preparation kit (Bioline, Menphis, TN, USA) following the manufacturer’s instructions. Libraries were sized and quantified using a 2200 TapeStation (Agilent Technologies, Santa Clara, CA, USA) and Qubit 3.0 fluorometer (Thermo Fisher Scientific, Waltham, MA, USA), then equimolarly pooled and sequenced using 2 × 150 bp reads on an Illumina NovaSeq 6000 S4 flow cell, aiming for 40× coverage of the ~282 Mb genome (Illumina Inc., San Diego, CA, USA). High-quality sequencing data were obtained for both replicates of each strain, except for G4 where only one replicate was successful.

The resulting sequence data were queried to obtain DNA sequences for the eight microsatellite loci from the seven phylloxera genotypes, to determine the absolute base pair sizes of alleles. First, the genomic coordinates of each locus were identified by aligning the forward and reverse primers for each locus to the *D. vitifoliae* v4.0 reference genome [26] using BLAST v2.12 [39]. Then, Tandem Repeats Finder v4.09 [40] was used to identify repeat units between those coordinates. Sequence reads were quality-filtered with fastp v0.20.0 [41] to only retain reads with a mean base quality >20, >50 bp in length, and containing <5 consecutive N bases, as well as to remove all adapter sequences and polyG tails which can occur in NovaSeq data [42]. The filtered reads were then mapped to the *D. vitifoliae* v4.0 reference genome [26] using the insertion–deletion (indel)-sensitive BWA-MEM v0.7.17 algorithm [43], retaining only properly paired reads with a mapping quality >30. Genotyping of the SSR repeat units from the genomic sequence data was conducted using HipSTR v0.6.2, a Hidden Markov Model (HMM)-based method which accounts for PCR stutter effects [44]. The HipSTR stutter model was estimated de novo for each locus, with genomic data from an additional 18 global population samples [26] included to increase the accuracy of the model estimation, but not used for further analysis. SSR genotype calls were then filtered to retain only those with a posterior probability over 90%, with no more than 15% of reads containing either a stutter artifact or a flanking sequence insertion–deletion, at least two reads spanning the SSR region, and with no evidence of allele or strand bias as determined by Fisher’s exact test (*p*-value > 0.01). The flanking regions of each SSR locus were separately genotyped using HaplotypeCaller in GATK v4.1.9 [45] and filtered to only retain calls with quality >30 and quality by depth >2. The resulting filtered VCF files from HipSTR and GATK were combined into a phased set of variants, and a sequence was generated for each allele using bcftools v1.15 [46]. For DVIT4, a flanking indel interfered with the HMM assembly used by HipSTR, so both the SSR and the flanking regions for this locus were genotyped with GATK HaplotypeCaller only.

Allele sequences obtained for each strain, along with the sequence present within the reference genome, were aligned to the original published sequences for each locus from Corrie et al. [8] (DVIT1 to DVIT4, GenBank accessions: AY056815-AY056818), Umina et al. [10] (DVIT5, DVIT6, GenBank accessions: AY371193, AJ969129), and Lin et al. [23] (DDVSSR3, DVSSR4, GenBank accessions: DQ016973, DQ016974) using the DECIPHER v2.24 package [47] in R 4.2 [48]. The allele sequences generated through HTS in the present study have been accessioned on GenBank, PQ788553–PQ788586.

## 3. Results

### 3.1. Platform 1: Polyacrylamide Genotyping

Examples of polyacrylamide gel autoradiographs for two of the SSR markers used in the current study (Dvit4 and Dvit5) are presented in Figure 1. These show typical allelic variation obtained when using the polyacrylamide genotyping platform. Alleles of the other Dvit microsatellite loci not illustrated here generally appear similar to Dvit4, while the Dvit5 locus produces numerous stutter bands (Figure 1).

### 3.2. Platform 2: Capillary Genotyping

The seven phylloxera genotypes examined in the current study represent greater than three quarters (24 of 31) of the Dvit locus alleles known from Australia (Table 1, Appendix A). However, in the current study, capillary allele sizes for these genotypes were found to vary by up to 3 bp in size from expected sizes described through previous polyacrylamide (“Platform 1”) genotyping (Table 1 and Table 2). The fluorescent allele sizes were found to most closely match the underlying DNA sequence bp sizes (derived from genome sequence data below, Table 1), only requiring the removal of the size of the universal primer tails to match these expected SSR allele sizes to within +/− 1 bp (Table 2). An example of genotyping results obtained in the present study using fluorescent capillary genotyping is presented in Figure 2.

### 3.3. Platform 3: HTS Genotyping

To verify the accuracy of the capillary allele sizes as well as the published genotype sizes, the target SSR regions were genotyped using high-coverage HTS sequence data from the seven target phylloxera strains (Figure 3, Appendix A). All strains were successfully HTS-genotyped for the Dvit1 and Dvit4 loci, while the G4 strain could not be HTS-genotyped for Dvit2 and Dvit3, and both G7 and G19 could not be HTS-genotyped for Dvit6 (Table 1). No strains were successfully HTS-genotyped for Dvit5, which consists of a simple poly-A repeat. For the two SSR loci from Lin et al. [23], DVSSR3 was successfully HTS-genotyped in only the G20, G30, G38, and G7 strains, while genotypes for the DVSSR4 locus were successfully obtained for all strains (Table 1). All HTS allele sequences were either strictly homozygous or heterozygous within strains, except for DVIT2 for G1 where one genotyped individual appeared homozygous at 256/256 bp, while the other was heterozygous 256/286 bp. Despite the mixed success of HTS genotyping for some strains and loci, concordance was generally seen between the allele sizes derived from the genomic dataset and those from the capillary genotyping protocol (Table 1 and Table 2).

### 3.4. Allele Size Comparisons Between Platforms

Comparing the allele sizes between the three platforms highlighted consistent locus-specific length differences associated with each technology. When the allele sequences genotyped from the HTS datasets were aligned to the original published reference sequences, it was apparent that these discrepancies were not due to variation in the SSR sequences alone, and insertion–deletions (indels) were also present in the flanking sequences (Figure 3, and Appendix A).

Dvit1 had a single deletion within the forward primer binding region, and a second deletion on the 3′ flank of the repeat sequence present across all sequenced strains (Figure 3, Appendix A). As the deletion was present in the primer binding region, PCR extension during amplification-based capillary genotyping would have incorporated the primer sequence, resulting in an extra base being added, with no overall length variation when compared to the polyacrylamide genotyping. However, this was not the case with the HTS data which did not use PCR amplification and thus detected both deletions, resulting in a −2 bp discrepancy compared to the original polyacrylamide platform (Figure 3, Table 1 and Table 2).

Dvit2 had three individual −1 bp deletions on the 5′ flank of the SSR sequence present across all sequenced strains, which accounted for the consistent −3 bp discrepancy between both the capillary and HTS data compared with the original published sequence and polyacrylamide allele lengths (Figure 3, Table 1 and Table 2).

There was a −2 bp discrepancy between both the HTS and capillary allele sizes and the original published sizes for Dvit3 (Figure 3, Table 1 and Table 2); however, only a single −1 bp deletion on the 5′ flank of the SSR sequence was found when the HTS sequences were aligned to the reference sequences (Figure 3), leaving a −1 bp discrepancy compared with the polyacrylamide allele sizing that could not be explained by indels in the flanking sequence.

For Dvit4, a +2 bp insertion on the 3′ flank of the SSR sequence (Appendix A), in addition to variation in the SSR itself, contributes to the longer alleles in strains G7, G19, G20, and G30, but was not present in the other strains (Appendix A). However, this indel does not explain the consistent −2 bp discrepancy between the HTS and polyacrylamide data, or the −3 bp discrepancy between the capillary and polyacrylamide data (Table 1 and Table 2). The original GenBank sequence for Dvit4 (AY056818.1) did not appear to be observed in the original polyacrylamide scoring [8]. However, in the present study, this DNA sequence was found to exactly match the 157 bp sized allele found in the G4 strain (historically called the ‘159’ allele). Therefore, the −2 bp size difference observed on the other genotyping platforms tested here can be explained by an error in the original polyacrylamide size scoring. An alternative explanation is required to explain the remaining approx. −1 bp difference between the capillary and HTS platforms.

For Dvit5, there was a −3 bp discrepancy between the capillary and polyacrylamide scoring (Figure 3, Table 1 and Table 2). However, this poly-A repeat could not be genotyped with the HTS data, so potential flanking indels or errors in the number of poly-As that likely explain this discrepancy could not be determined.

For Dvit6, there was a +2 bp insertion, as well as a separate −1 bp deletion on the 5′ side of the SSR sequence, that was consistently observed across all sequenced strains (Figure 3). Combined, these indels could explain the +1 bp difference compared with the polyacrylamide lengths; however, they do not explain the remaining approx. 1 bp discrepancy seen between the capillary and polyacrylamide allele lengths (Table 1 and Table 2). Similar to Dvit4, the allele lengths of the published Dvit6 sequence (AJ969129.1; 209 bp) did not match any of the allele sizes observed in the polyacrylamide data from Umina et al. [10]. However, in this case, the published Dvit6 reference sequence was generated through a separate study [9], with no information on the strain of origin, so it is unclear where this discrepancy was introduced.

Finally, for DVSSR3 and DVSSR4, there were no insertion–deletions found in either flank, with allele size differences between strains originating from the SSR itself (Figure 3, Table 1 and Table 2). For both these loci, the HTS and capillary measurements were in complete agreement. Note that there were no previous Australian phylloxera polyacrylamide measurements to compare for these two loci.

## 4. Discussion

Genotyping methods used for screening phylloxera SSR loci differ across laboratories worldwide [1]. The current preferred method for genotyping microsatellites is capillary fluorescent screening [29], which has been used, but so far not universally adopted, for phylloxera genotyping. Most loci are currently amplified individually (i.e., as single PCR reactions), rather than in multiplex, which involves more time, reagent and genotyping costs. In the present study, we address these issues by providing new laboratory methods for the amplification of phylloxera SSR markers via multiplex PCR, and genotyping using fluorescent labels on a capillary platform, greatly improving the efficiency and accuracy of phylloxera genotyping. We have demonstrated that this approach closely matches allele sizes obtained from HTS genomic data.

Multiplex PCR greatly reduces the number of PCR assays required for phylloxera genotyping. Screening via multiplex PCR is known to increase genotyping efficiency [29] and has become increasingly feasible through the widespread use of commercial multiplex kits that employ hot start enzymes [49]. In the present study, microsatellite SSR markers were successfully amplified via multiplex PCR reactions using previously developed phylloxera SSR primers [8,10,23], and screened using fluorescently labelled universal primers (*sensu* Blacket et al. [35]). PCR amplifications were performed in three multiplex reactions and one single reaction. Ideally, it would be preferable to reduce these reactions even further; however, combining loci into fewer reactions proved difficult with multiplex trials. We found that combinations of loci involving Dvit5 generally generated strong primer dimers, while some loci, Dvit4 and Dvit5, required relatively low annealing temperatures to avoid null alleles compared with the other SSR loci. Potentially greater genotyping efficiency could be achieved through poolplexing all PCR reactions together into a single pool prior to capillary separation, with fluorescent labels specifically chosen to allow for this (Appendix A). Although none of the loci tested here have alleles that are known to be identical in size, for ease and accuracy of genotyping, we found it preferable to genotype individuals as two poolplexed reactions (Figure 2). This reduced any potential error caused through fluorescent dye ‘pull-up/bleed-through’ [50] for pairs of loci whose allele size range overlaps in size (e.g., Dvit1 and Dvit5, Dvit2 and DVSSR3).

Populations of grape phylloxera worldwide appear to be highly genetically variable, and studies employing a relatively small number of microsatellites (i.e., six SSRs) have previously identified significant asexual genetic diversity outside the species native range, with more than 200 genotypes from Europe [9,51] and 83 from Australia [10]. Our study increased the set of SSR markers to eight for more precise detection of genetic variation, while allowing backwards compatibility with previous work. The addition here of DVSSR3 and DVSSR4, should prove valuable in improving genotype identification by increasing the number of diagnostic SSR differences, with alleles of these two loci also potentially associated with geographic regions [23], which may be useful for phylogeographic studies identifying potential sources of introduced phylloxera populations. While Umina et al. [10] found that increasing the number of loci from four to six SSR markers increased the number of phylloxera genotypes detected by approximately a third, this did not lead to detection of additional variation within the most viticulturally important common genotypes in Australia, i.e., G1 and G4 [10]. Increasing the number of markers to eight might enable differentiating additional genotypes within introduced populations.

Most SSR markers chosen for this study have multiple alleles at relatively high frequencies (Appendix A). While some of the markers (e.g., Dvit3) are not particularly variable (Table 1, Figure 2, Appendix A), they have proven useful in distinguishing biologically important genotypes of phylloxera (such as G1 and G4 in Australia). The SSR markers selected for this assay have been widely applied to the identification of phylloxera genotypes and mostly consist of variable di- and tri-nucleotide repeats (Appendix A), possessing allelic variation generally consistent with the standard stepwise slippage mutation model [52] (Levinson and Gutman, 1987), which produce highly reproducible genotypes (e.g., Umina et al. [10]).

The exception to this model is the SSR locus Dvit5, which consists of a simple poly-A repeat [9,10]. Single-base repeat markers are known to primarily vary through single-base mutations, which can result in difficulty in generating reliable genotypes [53]. However, this marker has previously proven particularly useful for identification of Australian genotypes [10] and was also found to amplify reliably in the present study (Figure 2). Generally, it is difficult to accurately score highly repetitive DNA sequences (such as SSR markers) using any DNA sequencing technology, as amplification errors are often generated through highly repetitive regions, resulting in SSR “stutter” bands [37]. While SSR genotyping from the genomic data did not provide complete coverage, genotypes could still be obtained for seven of eight loci. This allowed confirmation of the actual allele sizes (and sequences, submitted to GenBank in this study) across many of the target strains. The failure to genotype the Dvit5 locus through HTS genotyping in the current study is likely related to polymerase slippage and post-homopolymer error, which makes these forms of single-base expansions challenging to genotype with HTS sequencing data [54]. Furthermore, the mate-pair of some reads aligning to the flanking sequence of Dvit5 also aligned to more distant regions of the contig than would be expected by the insert size of the sequencing libraries, or separate contigs altogether. This suggests this locus may also represent a poorly assembled segment of the reference genome. The occurrence of amplification stuttering during various steps of the genomic sequencing protocol that involve amplification may also explain other discrepancies observed between platforms, for example, Dvit3, where HTS genotyping has likely preferentially sequenced a stutter band, resulting in an allele (171 bp) two bp (one repeat length) shorter than expected for genotype G30 (Table 1).

Calibrating genotyping systems to ensure universal scoring of allele sizes and data compatibility between laboratories and/or systems has previously been required due to incremental size differences between platforms [55], or the use of specific fluorescent dyes [32]. To ensure allele sizes were in line with previous polyacrylamide work, the capillary fragment sizes obtained from Geneious required two adjustments. The first was removal of the length of the forward primer tails (i.e., minus 15 to 18 bp, Table 1, *sensu* Blacket et al. [35]). The second was application of an adjustment factor to most of the Dvit loci due to apparent size shifts when compared to previous published allele sizes, requiring the addition of up to 3 bp to capillary genotype allele sizes to match the historic allele size data (Appendix A). These adjustments to the allele sizes were obtained from Geneious-generated fluorescent alleles at the expected sizes (*sensu* Umina et al. [10]) for Dvit1 to Dvit6, to within +/− 1 bp (Table 1 and Table 2). No adjustments were required for the two DVSSR loci, apart from a −1 bp allele name change for the DVSSR4 allele, occurring at 251 bp as ‘250’, to match the original size range reported in Lin et al. [23]. Most of the differences between the original and new genotyping platforms could be accounted for due to indel differences from the original clone sequences (Table 2). Each SSR locus was previously only represented on GenBank by a single cloned sequence. The HTS allele sequences generated in the present study now provide multiple sequences for most loci and allow characterisation of indel variation between alleles. One of the most notable findings of our study was that genomic data showed widespread insertion–deletion mutations in the flanking sequences of four loci (Dvit1, Dvit2, Dvit4, Dvit6), separate from the SSR sequences themselves.

When comparing the allele sizes among different genotyping platforms, it is important to consider that variation in amplified microsatellite allele sizes is affected by these flanking insertion–deletions, which can make the reported allele size appear out of phase with the expected length of the repeat unit (i.e., Table 2). While mutations in flanking regions are believed to evolve more slowly than the number of repeats within the SSR itself, flanking regions, as seen here, can exhibit significant polymorphism, which may be a serious concern when only fragment length analysis is performed [56]. For instance, when multiple flanking indels occur that match the length of the repeat motif itself, they can cause alleles to appear identical in state (i.e., have identical size) but are not identical by descent due to the size difference arising from a separate mutational process, a phenomenon known as size homoplasy [57]. The appearance of these flanking insertion–deletions in contemporary samples genotyped here compared to the original published allele sequences could arise either through errors in the original reference sequences, or possibly, through mutations that have occurred in the target strains since the original studies. The latter appears less likely, as flanking insertion–deletion mutations were generally found across all the sequenced strains. An example of the former is an insertion–deletion mutation appearing between two ambiguous ‘N’ bases in the original published allele sequence for DVIT6, suggesting this apparent indel when aligning sequences is more likely related to poor base calling from the original Sanger sequence trace. Indeed, errors in the original Sanger sequences are also present for Dvit3 and Dvit4 and are likely present (but could not be confirmed) for Dvit5 (Table 2). These sequence errors causing indels explain much of the apparent size differences observed between genotyping platforms (Table 2).

The remaining differences between allele sizes generated through capillary and HTS platforms (approx. 1 bp) that we are unable to explain through examination of sequence indels (Table 2) might potentially be due to fluorescent dye-specific effects, such as dye shift [32] or incremental size shifts caused by capillary allele size being estimated against size standards [55,58] rather than directly measured as discrete base pairs, as occurs in DNA sequencing. There were no consistent incremental differences observed relating to overall locus size in the present study, and so dye shift may be more likely for the unexpected allele size differences observed here (but has not been specifically tested). Notably, Dvit4 and Dvit6, which both exhibited unexplained allele size differences, employed the same FAM fluorophore.

In addition to providing information on the actual allele sizes and sequences across the target strains, the genomic datasets revealed mismatches between some of the primer sequences and binding sites for some loci. In particular, the T base situated 7 bp into the 5′ end of the Dvit1 forward primer sequence was absent across all the target strains, which might affect amplification efficiency as well as shift the resulting allele sizes. The T base situated 2 bp from the 3′ end of the Dvit2 forward primer was actually heterozygous for T/A in some genotypes, and similarly, the Dvit4 forward primer had a T > C transversion at the first base of the 5′ end of the forward primer, followed by a C > T transversion at the second base. While no genetic data were successfully obtained across any genotypes for the Dvit5 locus, when compared to the reference genome, there was a C > G transition 6 bp into the 5′ end of the forward primer. Finally, for the DVSSR4 primers, there was a single T > A transition at the first base of the 5′ end of the forward primer. However, unlike the deletion in Dvit1 primer binding sites, these single-base substitutions would not affect the resulting allele sizes but may reduce amplification efficiency, and this dataset offers an opportunity to further optimise primers for future studies.

## 5. Conclusions

The past two decades have seen genotypic identification of grape phylloxera become an integral part of control of this pest [10], informing the selection of resistant rootstocks and implementation of quarantine zones to prevent the spread of highly virulent genotypes. Our study validated eight SSR loci as standard markers for use in phylloxera genotype identification within Australia. Adoption of the standardised set of SSR markers and reference genotypes presented here should help with ensuring that molecular diagnostic results are consistent between different research groups and laboratories, to maintain compatibility of future work with valuable historic datasets, as previously performed for grapevine genotyping [59]. Presently, capillary genotyping provides the most reliable method for genotyping phylloxera, with the methods presented here being capable of genotyping all life stages of phylloxera, including single eggs. The capillary platform was the only new approach tested which could obtain complete genotypic data from all SSR markers (i.e., including Dvit5) in the present study. However, molecular diagnostics and phylogeographic studies are generally moving towards approaches other than SSR markers, such as genome-wide sequencing and SNP-based analyses. In our study, we have demonstrated that it is also possible to obtain SSR genotypes for most of these standard markers using HTS data, providing both assessments of both SNP variation and SSR allele sizes in the same assay. HTS genomic methods will no doubt continue to develop and potentially provide a new approach to examine phylloxera variation in even greater detail in the future, while allowing backwards compatibility of data with previous work.

## Figures and Tables

**Figure 1 insects-16-00230-f001:**
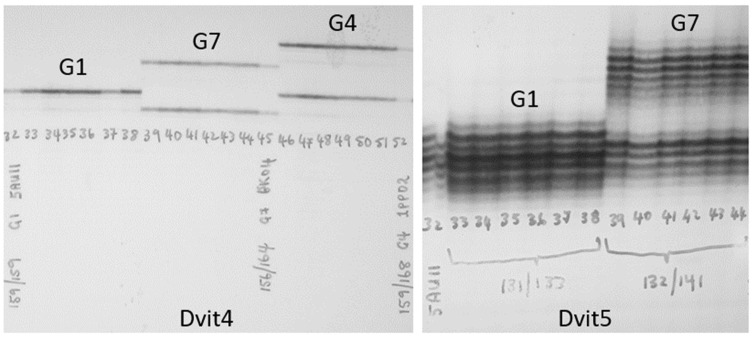
Polyacrylamide gel autoradiograph, showing Dvit4 (**left**) and Dvit5 (**right**) allelic variation observed in Australian phylloxera.

**Figure 2 insects-16-00230-f002:**
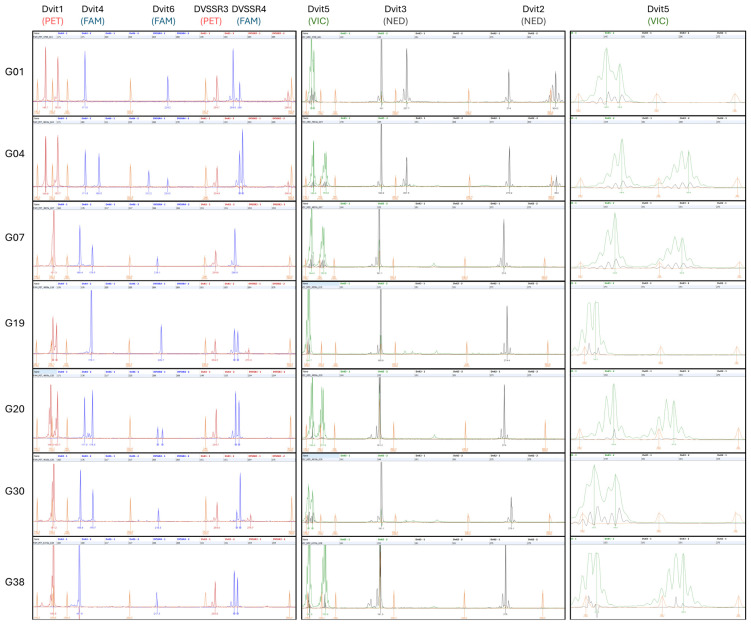
Co-amplification of eight phylloxera microsatellite loci from seven Australian genotypes, by multiplex PCR with tailed-forward and fluorescently labelled universal primers, screened through fluorescent genotyping on a capillary platform. Locus names, and fluorophores, indicated at top. Right images show greater detail of Dvit5 alleles, which are also shown in central pane. Note, allele sizes shown are unadjusted, and include primer tails of between 15 and 18 bp (see Appendix A).

**Figure 3 insects-16-00230-f003:**
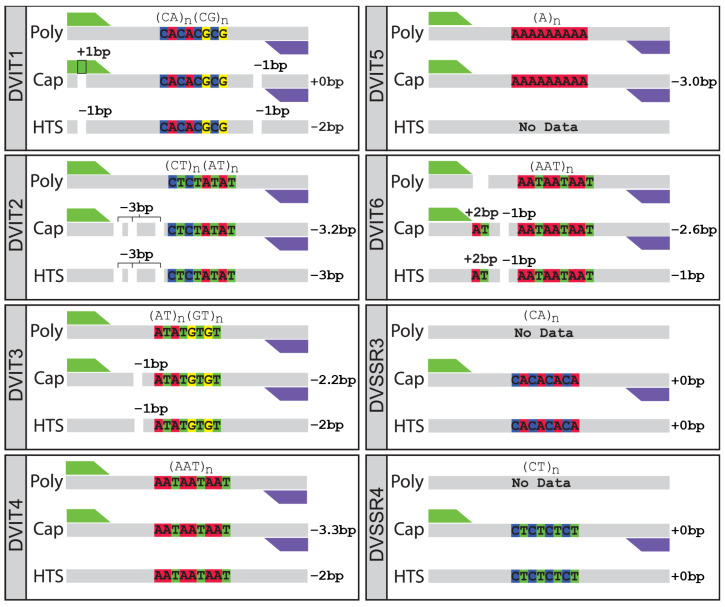
Comparison of alleles sizes scored between the three genotyping platforms in Australian grape phylloxera: Poly = polyacrylamide; Cap = capillary; and HTS = genomic sequencing. Green and purple trapezoids represent forward and reverse primers, respectively.

**Table 1 insects-16-00230-t001:** Microsatellite allele sizes for seven genotypes of grape phylloxera from Australia based on three different genotyping platforms. Allele sizes in base pairs. ND = not determined. White and grey shading indicates a genotype scored using three platforms.

	Genotype	Dvit1	Dvit2	Dvit3	Dvit4	Dvit5	Dvit6	DVSSR3	DVSSR4
Platform 1 (Original)	G1	128/136	259/289	175/190	159/159	131/133	211/211	237/280	248/253
Platform 2 (Capillary)	G1	(128.4/136.0)	(255.7/285.8)	(172.5/189.3)	(155.7/155.7)	(128.3/130.1)	(208.4/208.4)	(237.2/281.4)	(249.1/253.3)
Platform 3 (HTS)	G1	[126/134]	[256/256/286]	[173/189]	[157/157]	[ND]	[210/210]	[ND]	[249/253]
Platform 1 (Original)	G4	128/136	259/289	175/190	159/168	133/142	199/211	237/280	253/255
Platform 2 (Capillary)	G4	(128.5/136.1)	(255.7/285.7)	(172.4/189.2)	(155.9/164.7)	(130.0/138.5)	(196.3/208.4)	(237.2/281.3)	(253.2/255.1)
Platform 3 (HTS)	G4	[126/134]	[ND]	[ND]	[157/166]	[ND]	[198/210]	[ND]	[253/255]
Platform 1 (Original)	G7	134/134	259/259	175/175	156/164	132/141	205/205	237/237	250/250
Platform 2 (Capillary)	G7	(133.8/133.8)	(255.8/255.8)	(172.4/172.4)	(152.7/160.8)	(129.1/137.5)	(202.4/202.4)	(237.3/237.3)	(251.2/251.2)
Platform 3 (HTS)	G7	[132/132]	[256/256]	[173/173]	[154/162]	[ND]	[ND]	[237/237]	[251/251]
Platform 1 (Original)	G19	134/136	259/259	175/175	164/164	129/129	208/208	237/258	250/253
Platform 2 (Capillary)	G19	(133.8/136.1)	(255.8/255.8)	(172.5/172.5)	(160.8/160.8)	(126.3/126.3)	(205.4/205.4)	(237.3/258.2)	(251.3/253.4)
Platform 3 (HTS)	G19	[132/134]	[256/256]	[173/173]	[162/162]	[ND]	[ND]	[ND]	[251/253]
Platform 1 (Original)	G20	132/136	259/259	175/175	159/164	132/140	205/208	237/237	250/253
Platform 2 (Capillary)	G20	(131.8/136.1)	(255.9/255.9)	(172.6/172.6)	(155.7/160.8)	(129.0/136.5)	(202.4/205.4)	(237.2/237.2)	(251.2/253.3)
Platform 3 (HTS)	G20	[130/134]	[256/256]	[173/173]	[157/162]	[ND]	[204/207]	[237/237]	[251/253]
Platform 1 (Original)	G30	134/134	261/261	175/175	156/164	129/132	205/205	237/258	253/253
Platform 2 (Capillary)	G30	(133.7/133.7)	(257.8/257.8)	(172.5/172.5)	(152.6/160.7)	(126.3/129.2)	(205.4/202.4)	(237.1/258.1)	(253.2/253.2)
Platform 3 (HTS)	G30	[132/132]	[258/258]	[171/173]	[154/162]	[ND]	[204/204]	[237/259]	[253/253]
Platform 1 (Original)	G38	134/134	261/261	175/175	156/156	129/141	205/205	237/237	250/253
Platform 2 (Capillary)	G38	(133.8/133.8)	(257.8/257.8)	(172.6/172.6)	(152.8/152.8)	(126.3/137.5)	(202.5/202.5)	(237.3/237.3)	(251.3/253.4)
Platform 3 (HTS)	G38	[132/132]	[258/258]	[173/173]	[154/154]	[ND]	[204/204]	[237/237]	[251/253]

**Table 2 insects-16-00230-t002:** Comparison of average microsatellite allele sizes scored using three genotyping platforms (from Table 1 data). Allele size differences observed in platform comparisons are shown in base pairs (bp); n = number of alleles; Av = average; (±) = standard deviation; ND = not determined; N/A = not applicable.

Locus	Alleles (n)	Platforms 1 and 2	Platforms 1 and 3	Platforms 2 and 3	Explanation for Allele Size (Whole bp) Discrepancies
Dvit1	4	+0.0 bp	−2.0 bp	−2.0 bp	Yes, −2 bp deletions detected.
Dvit2	3	−3.2 bp	−3.0 bp	+0.2 bp	Yes, −3 bp deletions detected.
Dvit3	2	−2.2 bp	−2.1 bp	+0.3 bp	Partial, −1 bp deletion detected, −1 bp unaccounted for.
Dvit4	4	−3.3 bp	−2.0 bp	+1.3 bp	Partial, −2 bp due to error in original polyacrylamide scoring, capillary −1 bp unaccounted for.
Dvit5	7	−3.0 bp	ND	ND	Undetermined, alleles could not be sequenced.
Dvit6	4	−2.6 bp	−1.0 bp	+1.6 bp	Partial, overall indels detected result in +1 bp, capillary +1 bp unaccounted for.
DVSSR3	2	N/A	N/A	−0.1 bp	Yes, no discrepancy.
DVSSR4	4	N/A	N/A	−0.2 bp	Yes, no discrepancy.
	Total = 30	Av: −2.4 bp (±1.2)	Av: −2.0 bp (±0.7)	Av: +0.2 bp (±1.2)	Note: closest allele size matches occur between Platforms 2 and 3.

## Data Availability

All relevant data are within the manuscript. The SSR sequences produced in this study are available from the NCBI database (accession numbers PQ788553–PQ788586).

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
