# Peer review of "Bridging the Gap Between Platforms: Comparing Grape Phylloxera Daktulosphaira vitifoliae (Fitch) Microsatellite Allele Size and DNA Sequence Variation"

_insects, 2025, doi:10.3390/insects16020230_

Round 1

Reviewer 1 Report

Comments and Suggestions for Authors

This paper addresses currently used lab methods to visualize SSRs fragments applied to genotype D. vitifoliae lineages.  High sequencing defined Australien lineages are employed to verify and adjusted the genotyping profiles. The authors present a large and convincing dataset. 

Since SSR Genotyping has been used in the last decade in population studies of D. vit. and the genotyping of single individuals a vast amount of information on this strains are available worldwide. Hence the authors tried to connect these Allele/combinations with state of the Art sequencing techniques. 

The specimen and samples employed are from Australien origin mainly, as are the marker sets used (there are different sets applied in the rest of the world). Thus, the title of the paper should clearly state that this fact. Eg: 

Bridging the gap between platforms: Comparing grape phylloxera Daktulosphaira vitifoliae (Fitch) from Australia microsatellite allele size and DNA sequence variation

The paper (as a methods paper) provides lots of details in the result section which are less relevant for the type of work presented. Furthermore I was missing a more conclusive Outlook on how to use  the Sequencing methods (automated) for the Phylloxera work. I disagree with the authors, the SSR system as far as I am concerned already outdated. This would also increase the novelty of the contents - which are in this state rather low. 

There are some minor corrections (mainly nomenclature)  needed in the introduction pertaining the Biology of Phylloxera. The figure (of gel SSRs) could be omitted (or must include a DNA marker reference).  

The authors discuss to a minor extent the occurrence of null alleles which have been reported from other groups. I would expect this to be strengthened. I also would have suggested to include further samples with known genetic background (sexual/asexual). However I am aware that such samples are not easy to produce in labs. As far as I can see the references available are being cited and discussed. 

Taken together, given the suggestion I would  accept the paper with major revisions. 

Author Response

Reviewer 1

Comment 1: The specimen and samples employed are from Australian origin mainly, as are the marker sets used (there are different sets applied in the rest of the world). Thus, the title of the paper should clearly state that this fact. Eg: Bridging the gap between platforms: Comparing grape phylloxera Daktulosphaira vitifoliae (Fitch) from Australia microsatellite allele size and DNA sequence variation

Response 1: The scope of the study is broader than Australia, so we would prefer to not include Australia specifically in the title. For clarity we do specify numerous times throughout that Australian samples were used for this study, including in the Abstract, and in many other sections of the paper.

Comment 2: The paper (as a methods paper) provides lots of details in the result section which are less relevant for the type of work presented. Furthermore I was missing a more conclusive Outlook on how to use the Sequencing methods (automated) for the Phylloxera work. I disagree with the authors, the SSR system as far as I am concerned already outdated. This would also increase the novelty of the contents - which are in this state rather low.

Response 2: SSR genotyping using size separation (e.g. Capillary fluorescent method presented here) is still currently the only method that can identify phylloxera strains for comparison with historical datasets. We have now specifically stated this in the Conclusion (lines 479-481). This is due to genome sequencing being unable to generate accurate sequence data from low complexity SSR alleles, such as Dvit5, as demonstrated in the current study (see lines 387-393, Tables 1 & 2, etc.).

Comment 3a: There are some minor corrections (mainly nomenclature) needed in the introduction pertaining the Biology of Phylloxera.

Response 3a: The specific changes requested have not been specified, so we are unsure which corrections are required? The name of phylloxera as used “Daktulosphaira vitifoliae (Fitch)” is the most commonly employed name worldwide, (see the Global Biodiversity Information Facility, GBIF, https://www.gbif.org/species/2049512). As is the name for European Grapevine, “Vitis vinifera L.”, (see GBIF https://www.gbif.org/species/144100931).

Comment 3b: The figure (of gel SSRs) could be omitted (or must include a DNA marker reference). 

Response 3b: The primary intention of this Figure was to show what typical alleles look like using the polyacrylamide system, this being the original of the three genotyping platforms compared. We think that it is important to include this Figure in the paper for direct comparison with the newer approaches. Further explanation is provided in related “Comment 5” (below).

Comment 4: The authors discuss to a minor extent the occurrence of null alleles which have been reported from other groups. I would expect this to be strengthened. I also would have suggested to include further samples with known genetic background (sexual/asexual). However I am aware that such samples are not easy to produce in labs. As far as I can see the references available are being cited and discussed.

Response 4: Sexual reproduction has not been demonstrated in Australia, with all field collected and laboratory phylloxera thought to be asexual. This has been previously documented in the literature, and is highlighted in the text (lines 52-58).

Reviewer 2 Report

Comments and Suggestions for Authors

Grape phylloxera, Daktulosphaira vitifoliae (Fitch), is one of the most serious pests of grapevines worldwide. Identification of phylloxera genotypes is important for successful pest management. The current study presents new laboratory methods for amplifying a standard set of eight phylloxera microsatellite markers using PCR incorporated fluorescently labelled primers, and genotyped on an ABI capillary platform. Comparison of allele size data scored on polyacrylamide, capillary, and high throughput sequencing platforms. Results revealed that the capillary genotyping most closely matched the HTS allele sizes. the new method which simultaneously screen multiple microsatellites, can reduce the cost, increase the high throughput capacity of this molecular approach, and make phylloxera population genetic studies more accessible.

1. In the section of Results: In Figures 1,only two of the SSR markers (Dvit4 and Dvit5) are presented on polyacrylamide gel autoradiographs in this manuscript. How about the results of the other Dvit loci such as Dvit1, 2,3…? Why not present here? Could you provide an explanation or I recommended to make supplement?

2. According to the research here,the new method of amplification of  phylloxera SSR markers via multiplex PCR and genotyping using fluorescent labels on a capillary platform, results demonstrated closely matches with that from HTS genomic data, thus improving the efficiency and accuracy of phylloxera genotyping. What is the reason?Why the multiplex PCR is better?I Suggest adding a description in the discussion section.

3. What is the disadvantage of optimised capillary genotyping method presented in your work? Could you point out what improvements it needs in the future work. And Can it replace the HTS platform in genotyping according to your findings?

Comments on the Quality of English Language

The writing of the article needs to be properly revised before publishing.

Author Response

Reviewer 2

Comment 5: In the section of Results: In Figures 1,only two of the SSR markers (Dvit4 and Dvit5) are presented on polyacrylamide gel autoradiographs in this manuscript. How about the results of the other Dvit loci such as Dvit1, 2,3…? Why not present here? Could you provide an explanation or I recommended to make supplement?

Response 5: These were the only polyacrylamide images readily available to the authors. The two markers shown in the manuscript illustrate typical variation observed across all of the Dvit markers, when using the polyacrylamide system. Other Dvit markers produce alleles which look similar to the Dvit4 example, while the Dvit5 example shows extreme stuttering. We have changed the wording in the text to reflect this (see lines 210-214).

Comment 6: According to the research here,the new method of amplification of phylloxera SSR markers via multiplex PCR and genotyping using fluorescent labels on a capillary platform, results demonstrated closely matches with that from HTS genomic data, thus improving the efficiency and accuracy of phylloxera genotyping. What is the reason?Why the multiplex PCR is better?I Suggest adding a description in the discussion section.

Response 6: The increased accuracy differences are likely due to these methods being more accurate than the previous polyacrylamide genotyping method (as we have stated in the manuscript (lines 324-333).  See further notes on multiplex efficiency in related “Comment 7” (below)

Comment 7: What is the disadvantage of optimised capillary genotyping method presented in your work? Could you point out what improvements it needs in the future work. And Can it replace the HTS platform in genotyping according to your findings?

Response 7: The disadvantage of capillary genotyping is the labour / material cost required to genotype each sample is still relatively high, especially given lower genomic sequencing costs. The new methods presented here using multiplex PCR greatly reduce the number of PCR and genotyping reactions, and are thus more efficient than the older polyacrylamide method. The capillary system is also more accurate with allele sizes closely matching those generated from whole genome sequencing (when this data could be obtained). (See modified text, lines 324-333). 

Comment 8: Comments on the Quality of English Language

The writing of the article needs to be properly revised before publishing.

Response 8: We have revised the writing throughout the manuscript during the revision process (using Track Changes). Note, we have completely replaced the Simple Summary section.

Reviewer 3 Report

Comments and Suggestions for Authors

The manuscript presents a well-executed study with clear writing and logical flow. The conclusions are generally well-supported by the data presented. However, I have some concerns about the practical advantages of the proposed method in the current genomic era.

Major Comments:

  1. While the authors propose a novel PCR-based approach for low-cost, rapid genotyping in laboratory settings, the manuscript lacks a compelling demonstration of its advantages over modern sequencing technologies. Specifically:

    • The authors should provide a more detailed cost-benefit analysis comparing their method with current sequencing approaches
    • The efficiency gains in terms of time and labor need to be more clearly quantified
    • The accuracy and reliability of SNP calling compared to modern sequencing methods should be addressed
  2. The manuscript would benefit from:

    • A more comprehensive comparison of the data quality between their method and contemporary sequencing approaches
    • Discussion of specific scenarios where this method might be preferable to modern sequencing technologies
    • Clear examples of applications where rapid, low-cost genotyping would outweigh the benefits of more comprehensive genomic data

Author Response

Comment 9: While the authors propose a novel PCR-based approach for low-cost, rapid genotyping in laboratory settings, the manuscript lacks a compelling demonstration of its advantages over modern sequencing technologies. Specifically: The authors should provide a more detailed cost-benefit analysis comparing their method with current sequencing approaches. The efficiency gains in terms of time and labor need to be more clearly quantified. The accuracy and reliability of SNP calling compared to modern sequencing methods should be addressed.

Response 9: This has mostly been addressed above in “Comment 7”. Specifically, the only method capable of accurately genotyping all loci is the capillary method, some loci e.g. Dvit 5 could not be sequenced due to low complexity of the repeat (poly-A) region (see text lines 384-393). Performing the capillary genotyping using multiplex, has increased the efficiency, reducing the number of PCR and genotyping reactions required (see text lines 324-341).

Comment 10: A more comprehensive comparison of the data quality between their method and contemporary sequencing approaches. Discussion of specific scenarios where this method might be preferable to modern sequencing technologies. Clear examples of applications where rapid, low-cost genotyping would outweigh the benefits of more comprehensive genomic data.

Response 10: Genome sequencing cannot characterise all of the phylloxera microsatellite genotypes that have been previously defined based solely on SSR markers (see “Comment 2” above).

Reviewer 4 Report

Comments and Suggestions for Authors

The study confirms and standardizes SSR loci as markers for identifying phylloxera genotypes in Australia. It suggests that using these SSR markers and reference genotypes will help keep studies consistent and compatible with past phylloxera genotyping data. It also shows that the proposed capillary genotyping method is reliable, cost-effective, and helps improve local phylloxera management. In general, the article is clear, relevant for the Australian viticulture, well-structured, methodology is accurate and results are well discussed.

Manuscript rating:

Novelty: The article addresses a gap in phylloxera genotyping by ensuring good comparisons between new and historical data, as well as offering a more affordable alternative to non-multiplex SSR PCRs.

Scope: It fits the journal's scope.

Significance: The results are well interpreted, and the conclusions are supported by the data.

Quality: The article is well-written, the analyses are appropriate, and the results are solid. Only one figure needs adjustment to meet the quality requirements.

Scientific soundness: The study is well-designed, with enough detail in the methods, tools, software, and protocols to ensure reproducibility.

Interest to the readers: The conclusions are of broad interest, not only to those studying phylloxera in Australia but also to researchers worldwide, making the paper relevant to a wide readership.

Overall merit: This work has significant value, providing valid results that improve understanding of phylloxera genotyping.

English level: Native.

Overall recommendation:

Accept after minor revisions

Specific comments are separated in the different manuscript sections.

Introduction:

Line 48:

There’s evidence suggesting that V. vinifera has a Euroasian origen (e.g. This et al., 2006 or KarataÅŸ et al., 2014; Naquinezhad et al. 2018; Riaz et al. 2018 etc.). Regarding this same line, if the damage is only specified for V. vinifera it gives the impression that symptoms are different in other Vitis species.

Line 54:

I’d suggest removing the word “microscopic” since a trained eye can actually spot phylloxera with the naked eye.

Line 107: After centrifuging, the 2µl of template DNA used for each PCR reaction are pipetted from the supernatant, right? Better clarify.  

Line 189: I suggest sticking to SSR instead of STR.

Results:

Table 2: The title must explain what those bp mean. For example, for Dvit1 in the second comparison, that – 2 bp means that platform 1 detects 2 less bp than HTS? It must be specified in the title. What is the meaning of “minus original” and “minus capillary”? Is not a straight comparison between 1, 2 and 3 platforms?

The third column is not very clear. By "indels," do you mean insertions and deletions? In Dvit2, for example, the explanation for -2 bp refers to deletions, but what about the +0.2? Dvit4: Partial, -2 bp due to the original polyacrylamide scoring, capillary -1 bp (dye shift?). What about these? In my opinion, this column could be removed and explained in the discussion section, as these are explanations for the observed discrepancies. Another suggestion is to move the "note" from that table to the main text, as it is the core of these results.

Figure 2: The electropherograms are difficult to interpret due to its low resolution and the small font size, which makes it hard to read. I recommend improving the resolution and considering moving it to the supplementary materials for better accessibility and clarity.

Figure 3: Consider removing “A to H = SSR loci Dvit1, Dvit2, Dvit3, Dvit4, Dvit5, Dvit6, DVSSR3 and DVSSR4 respectively” since that evident and very clear in the figure (you could even remove the letters since panels are very clear). You could also clarify that the purple and the green trapezoids represent forward and reverse primers.

Discussion:

It would be helpful if you could provide an approximate estimate of the relative cost reduction.

Line 354: I suggest adding some references from the American continent.

Supplemental material:

Figure 2: It would be helpful to specify that each dot represents an observation, as this is not immediately self-explanatory and could benefit from clarification.

Figure 3-9. Please add reference to the bases colours.

Author Response

Introduction:

Comment 11: Line 48: There’s evidence suggesting that V. vinifera has a Euroasian origin (e.g. This et al., 2006 or KarataÅŸ et al., 2014; Naquinezhad et al. 2018; Riaz et al. 2018 etc.). Regarding this same line, if the damage is only specified for V. vinifera it gives the impression that symptoms are different in other Vitis species.

Response 11: This section has been re-worded in the text to clarify that we are referring to damage to Vitis vinifera (lines 45-49).

Comment 12: Line 54:

I’d suggest removing the word “microscopic” since a trained eye can actually spot phylloxera with the naked eye.

Response 12: We have now clarified that we are referring to laboratory diagnostics. Changing the wording to “Diagnostic laboratory identification of this serious pest species is currently achieved through visual microscopic examination”. Lines 49-51.

Comment 13: Line 107: After centrifuging, the 2µl of template DNA used for each PCR reaction are pipetted from the supernatant, right? Better clarify. 

Response 13: Done. Line 107.

Comment 14: Line 189: I suggest sticking to SSR instead of STR.

Response 14: Done. STR changed to SSR between lines 186-199.

Results:

Comment 15: Table 2: The title must explain what those bp mean. For example, for Dvit1 in the second comparison, that – 2 bp means that platform 1 detects 2 less bp than HTS? It must be specified in the title. What is the meaning of “minus original” and “minus capillary”? Is not a straight comparison between 1, 2 and 3 platforms?

Response 15: Corrections made to title of Table 2. Title text changed to “Alleles size differences observed in platform comparisons are shown in base pairs”. The notes on which platform was “minus” which have been removed from column headings.

Comment 16: The third column is not very clear. By "indels," do you mean insertions and deletions? In Dvit2, for example, the explanation for -2 bp refers to deletions, but what about the +0.2? Dvit4: Partial, -2 bp due to the original polyacrylamide scoring, capillary -1 bp (dye shift?). What about these? In my opinion, this column could be removed and explained in the discussion section, as these are explanations for the observed discrepancies. Another suggestion is to move the "note" from that table to the main text, as it is the core of these results.

Response 16: The title of the last column has been modified to “Explanation for allele size (whole bp) discrepancies”, to only cover the whole base pair differences observed. Indels have now been defined as “deletions, insertions, or indels” (i.e., multiple insertions and deletions), to match the discussion of each locus in the text.

Comment 17: Figure 2: The electropherograms are difficult to interpret due to its low resolution and the small font size, which makes it hard to read. I recommend improving the resolution and considering moving it to the supplementary materials for better accessibility and clarity.

Response 17: We have replaced the Figure with a higher resolution version, which allows details to be seen when zoomed in. We consider this figure to be critically important to the manuscript and so should be in the main text, as it demonstrates what typical capillary genotyping results look like. Which this genotyping platform being a major focus of the paper and new laboratory protocols presented.

Comment 18: Figure 3: Consider removing “A to H = SSR loci Dvit1, Dvit2, Dvit3, Dvit4, Dvit5, Dvit6, DVSSR3 and DVSSR4 respectively” since that evident and very clear in the figure (you could even remove the letters since panels are very clear). You could also clarify that the purple and the green trapezoids represent forward and reverse primers.

Response 18: We have removed labels A to H, and have now defined primer colours in the legend.

Discussion:

Comment 19: It would be helpful if you could provide an approximate estimate of the relative cost reduction.

Response 19: This has been covered above in multiple comments (i.e. fewer PCR and genotyping reactions).

Comment 20: Line 354: I suggest adding some references from the American continent.

Response 20: We have clarified that we are referring to detecting significant asexual genetic diversity in populations outside the species native range (line 355).

Supplemental material:

Comment 21: Figure 2: It would be helpful to specify that each dot represents an observation, as this is not immediately self-explanatory and could benefit from clarification.

Response 21: Done. Added “Each dot represents an allele present in a known genotype”. This change has also been added to the main text (lines 496-497)

Comment 22: Figure 3-9. Please add reference to the bases colours.

Response 22: Done. “DNA bases represented by G = yellow, A = red, T = green, C = blue”. The actual DNA sequences of alleles submitted to GenBank have now also been added as text, labelled their GenBank accession numbers. The GenBank accession numbers have also been added to the main text (line 207).

Round 2

Reviewer 3 Report

Comments and Suggestions for Authors

The authors have addressed the concerns I have raised, Now I dont have further questions.

Reviewer 4 Report

Comments and Suggestions for Authors

Paper is ok to be published.